# Effect of Electrohydrodynamic Drying on Drying Characteristics and Physicochemical Properties of Carrot

**DOI:** 10.3390/foods12234228

**Published:** 2023-11-23

**Authors:** Yanghong Wang, Changjiang Ding

**Affiliations:** College of Science, Inner Mongolia University of Technology, Hohhot 010051, China; 202110905076@imut.edu.cn

**Keywords:** electrohydrodynamics, carrot drying, fourier infrared spectrum, SEM, low-field NMR

## Abstract

This study investigates the effects of electrohydrodynamic (EHD) drying technology on the drying kinetics, microstructure, quality, and nutritional components of carrots, along with conducting experiments on EHD drying under different voltage gradients. The experimental results showed that EHD drying technology could significantly increase the drying rate and the effective moisture diffusion coefficient. Within a certain range, the drying rate was directly proportional to the voltage. When the range was exceeded, the increase in voltage had a minimal effect on the drying rate. In terms of quality, the EHD drying group’s color, shrinkage rate, and rehydration performance were superior to the control group, and different voltages had no significant effect on the shrinkage rate and rehydration performance. The retention of carotenoids in the EHD drying group was 1.58 to 2 times that of the control group. EHD drying had a negative impact on the total phenolic content and vitamin A content of dried carrot slices. Based on the results of infrared spectroscopy and scanning electron microscopy (SEM), the dehydrated carrot slices showed wrinkling due to water loss, with numerous pores, a generally intact structure, and retained functional groups. EHD drying had a significant impact on the secondary structure of proteins, where an increase in voltage led to an increase in disordered structure, with a smaller proportion of disordered structure in the lower voltage group compared to the control group, and a similar proportion of disordered structure between the higher voltage group and the control group. Results from low-field nuclear magnetic resonance (NMR) showed that EHD drying could retain more bound water compared to the control group, with the best retention of cellular bound water at a voltage of 26 kV and the best retention of cellular immobilized water at a voltage of 38 kV, indicating the superiority of EHD drying in preserving cellular structure. This study provided a theoretical basis and experimental foundation for the application of electrohydrodynamic drying technology to carrot drying, and promoted the practical application of EHD drying technology.

## 1. Introduction

Carrot is one of the top ten global vegetable crops and is widely popular due to its high nutritional value. Four phytochemicals found in carrots are phenolics, carotenoids, polyacetylenes, and vitamin C. These compounds possess antioxidant, anti-inflammatory, plasma lipid regulation, and anti-tumor properties, which help to reduce the risk of cancer and cardiovascular diseases [1].

Fresh carrots have high moisture content and are prone to decay and spoilage due to their high metabolic activity and respiration rate. Therefore, drying is commonly used to extend their shelf life and maintain nutritional quality. The most common drying method for carrots is natural drying, but it is limited by weather conditions, leading to the development of various drying techniques. Mohammadi et al. [2] studied the effects of radiation energy vacuum (REV) drying, freeze-drying, and air drying on the nutritional value and sensory quality of broccoli, orange, and carrot, and found that REV drying significantly reduced processing time and better retained nutrients such as vitamin C and β-carotene, while achieving the highest scores in all sensory characteristics among the radiation energy-dried products. Kocabiyik et al. [3] studied the infrared radiation drying of carrot slices and investigated the effects of process variables (infrared power, air velocity) on the drying time, specific energy consumption, and quality parameters (shrinkage rate, rehydration rate, and color) of dried carrots. The study found that drying rate increased with increasing infrared power and that shrinkage rate, rehydration rate, and color parameters were influenced by process variables. The study by Ignaczak et al. [4] showed that freeze-dried carrot samples had a high dry matter content and low water activity; microwave vacuum-dried samples had large color differences, while those dried by convection and freeze-drying had smaller color differences. Radiant Energy Vacuum (REV) dehydration was an advanced method of fast, low-temperature drying, but the equipment manufacturing and installation costs were high [2]. The advantages of freeze-drying were evident, but its application in the food industry was not widespread. This was because freeze-drying was time-consuming, and the equipment was expensive [5]. While ensuring drying quality, saving time, and energy costs were the development trends in the drying industry, research into new technologies was imperative. Amami et al. [6] studied the effects of pulsed electric field (PEF) pretreatment on the dehydration and rehydration properties of carrots; they demonstrated that PEF not only increased the water loss rate and solid content during carrot osmotic dehydration but also enhanced the rehydration ability of carrots, albeit to some extent reducing the hardness of the rehydrated product. Kamran et al. [7] studied the effects of PEF, ultrasound, and blanching pretreatments on the drying kinetics, energy consumption, and quality attributes of EHD dried apples. The results showed that only PEF pretreatment could reduce the energy consumption of the drying process, significantly decrease the drying time and energy consumption, but it could compromise the sensory appeal of the dried apples due to enzymatic browning. It can be seen that the combination of PEF pretreatment and EHD drying did not achieve ideal results. Zhou et al. [8] conducted a study on the plasma pretreatment of goji berries followed by hot air drying and found that plasma pretreatment significantly shortened the drying time, improved rehydration capability, and color quality, but the impact on the retention of plant chemical substances after drying varied with the duration of plasma treatment. Cold plasma (CP) technology was widely recognized as a very promising non-thermal drying pretreatment technique, with advantages in terms of being environmentally friendly, having low energy consumption, enhancing the drying rate, and reducing nutrient loss during the drying process. The combination of CP pretreatment with hot air drying resulted in reduced drying time and lowered energy consumption, but the equipment price of CP technology was high, and the process control of CP pretreatment remained a challenge [9]. These pretreatments solely used physical means to quickly change the material structure to accelerate drying, whereas electrohydrodynamic acted on the material for an extended duration to expedite drying.

The EHD drying technique is a novel drying method that utilizes electric field energy transfer for drying [10]. It features non-heating of the material [11], energy savings [12,13], low equipment cost, and it is particularly suitable for drying heat-sensitive materials. There have been initial studies exploring the application of the EHD drying technique for carrot drying. Ding et al. [14] conducted a comparative experiment between high-voltage electric field drying and hot air drying using carrots and found that the high-voltage electric field accelerated the drying process and effectively retained carotenoids. Based on the research by Ding et al., we added various detection methods, such as infrared spectroscopy and low-field nuclear magnetic resonance, to systematically analyze the dried samples and study the effects of EHD drying technology on the basic functional groups and protein structure of carrots, as well as the distribution of water in the dried samples. This study provides a more comprehensive theoretical and experimental basis for the application of EHD drying technology in carrot drying, further promoting its practical application.

## 2. Materials and Methods

### 2.1. Original Material

Fresh, disease-free carrots were purchased from a supermarket near Inner Mongolia University of Technology. The carrots were washed, dried, and cut into cylindrical shapes with a thickness of 2 mm and a diameter of 35 mm. During the preparation process, the center of the circular mold was aligned with the center of the circular carrot slices to maintain consistency.

### 2.2. Experimental Installation

As shown in Figure 1, the experimental setup consists of a needle-type electrode system, a 5 mm-thick medium plate, and a high-voltage power supply control system (YD(JZ)-1.5/50, Wuhan Boyu Electric Power Equipment Co., Ltd., Wuhan, China). The high-voltage power supply control system can output AC or DC voltage, with the AC voltage adjustable from 0 to 50 kV and the DC voltage adjustable from 0 to 70 kV. In this experiment, an AC voltage was used (the EHD drying effect of alternating current power supply is significantly greater than that of direct current power supply [15,16]). The needle-type electrode was connected to a high-power control system. The grounded electrode was a stainless steel plate with dimensions of 1000 mm × 550 mm, and the distance between the needle tip of the needle-type electrode and the grounding electrode was 80 mm. A 5 mm-thick medium plate was placed on the grounding electrode steel plate. The needle length was 20 mm, the needle diameter was 1 mm, and the distance between each needle was 40 mm. During the experiment, the room temperature was maintained at 26 ± 3 °C, and the humidity of the air was maintained at 25% RH.

### 2.3. Empirical Method

The prepared carrot slices were placed in an EHD drying system. The mass of each sample ranged between 2.0 g and 2.3 g. For each experiment, a certain amount of carrot slices were placed (in this experiment, 50 carrot samples were used, with three samples reserved for monitoring the drying process, color difference measurement, and the remaining samples for measuring shrinkage rate, rehydration rate, infrared spectrum, scanning electron microscopy, etc.). The voltage (0 kV, 20 kV, 26 kV, 32 kV, 38 kV, 44 kV, and 0 kV being the control group) was selected for the drying experiment. The mass of the carrot slices was measured every 45 min using an electronic balance (BS124S, Shanghai Guanglu Electronic Technology Co., Ltd., Shanghai, China), until the moisture content of the carrots fell below 1%, at which point drying was stopped. After drying, the sample mass ranged between 0.23 g and 0.26 g, with a moisture content below 1%.

### 2.4. Moisture Content

During drying, the moisture content of the carrots at certain times was measured, and the calculation formula [17] was as follows:(1)$$M_{t}=\frac{m_{t}-m_{s}}{m_{s}}$$
where $M_{t}$ is the dry basis moisture content, $m_{t}$ is the mass of the carrots at *t* time (g), and $m_{s}$ is the weight of the carrot after removing all of the moisture.

### 2.5. Drying Rate

The drying rate was represented by the change in dry basis moisture content between adjacent times during the same time interval, and the calculation formula [18] was as follows:(2)$$V_{t}=\frac{M_{t}-M_{t-1}}{t_{d}}$$
where $V_{t}$ is the drying rate (g/(g · min)); $M_{t-1}$ is the dry basis moisture content of the carrots at t−1 time; and $t_{d}$ is the time interval between *t* time and t−1 time (min).

### 2.6. Moisture Ratio

The moisture ratio (MR) [19] was calculated using the formula:(3)$$MR=\frac{M_{t}-M_{e}}{M_{0}-M_{e}}$$
where $M_{e}$ is the equilibrium dry basis moisture content of the carrots, and $M_{0}$ is the initial dry basis moisture content of the carrots. Compared to the initial moisture content ($M_{0}$) and the moisture content at time *t* ($M_{t}$), the moisture content of the carrots when drying is complete can be neglected, and the equation can be simplified as follows:(4)$$MR=\frac{M_{t}}{M_{0}}$$

### 2.7. Effective Moisture Diffusivity

The Fick’s diffusion equation can be used to describe the drying characteristics of biological products. For long drying times, MR<0.6, Fick’s diffusion equation can be simplified as follows [20]:(5)$$lnMR=ln\frac{8}{\pi^{2}}-\frac{\pi^{2}D_{eff}t}{4L_{0}^{2}}$$
where $D_{eff}$ is the effective moisture diffusivity of the sample, and $L_{0}$ is half of the sample thickness. Based on the experimental data, a straight line equation lnMR−t is plotted, and the slope of the straight line equation is denoted as $K=\frac{\pi^{2}D_{eff}}{4L_{0}^{2}}$. The value of $D_{eff}$ is calculated using the slope.

### 2.8. Color

We used the CIE Lab color space for color measurement. An automatic colorimeter (3nh-NR60CP, Shenzhen Threenh Technology Co., Ltd., Shenzhen, China) was used to measure the color of the samples, with the color of a white plate as the standard. For each sample, five points were taken, and the average values were calculated. The measured parameters included brightness (*L*), redness (*a*), yellowness (*b*), difference in color saturation (Δ*C*), difference in hue angle (Δ*α*°), and total color difference (Δ*E*) [21]. Each parameter was calculated using the following formulas:(6)$$\Delta C=\sqrt{a_{1}^{2}+b_{1}^{2}}-\sqrt{a_{0}^{2}+b_{0}^{2}}$$
(7)$$\Delta \alpha^{\circ }=tan^{-1}\left(\frac{b_{1}}{a_{1}}\right)-tan^{-1}\left(\frac{b_{0}}{a_{0}}\right)$$
(8)$$\Delta E=\sqrt{{\left(L_{1}-L_{0}\right)}^{2}+{\left(a_{1}-a_{0}\right)}^{2}+{\left(b_{1}-b_{0}\right)}^{2}}$$
where $L_{0},\;a_{0},\;b_{0}$ represent the measured value of fresh carrot; $L_{1},\;a_{1},\;b_{1}$ represent the carrot measured value after dry treatment.

### 2.9. Shrinkage

In the experiment, the shrinkage was measured using the displacement method. A piece of dried carrot was placed in a 25 mL graduated cylinder filled with 18 mL of water, and the rise of the liquid level was recorded for three parallel experiments. The shrinkage was expressed as the percentage by which the volume of the carrot decreased compared to its initial volume, and the calculation formula [22,23] was as follows:(9)$$S=(\frac{V_{0}-V}{V_{0}})\times 100\%$$
where *S* is the shrinkage percentage, $V_{0}$ is the initial volume of the carrot, and *V* is the volume of the carrot after drying.

### 2.10. Rehydration Performance

In the experiment, the constant temperature water bath was set to 37 °C (high temperatures can cause significant damage to plant tissue [24]). We added 100 mL of deionized water to three 150 mL cups and preheated them in a constant temperature 37 °C water bath. Then, we placed three dried carrot slices in each of the three cups and soaked them for 3 h. Afterward, we removed the carrot slices and completely absorbed the surface moisture with filter paper. The mass of the carrot slices before and after rehydration was measured using a Sartorius electronic scale (BS124S, Shanghai Guanglu Electronic Technology Co., Ltd., Shanghai, China). The rehydration rate was calculated using the following formula [25]:(10)$$RR=\frac{m_{a}}{m_{b}}\times 100\%$$
where RR is the rehydration rate of the carrot slices, $m_{a}$ is the mass of the carrot slices after rehydration (g), and $m_{b}$ is the mass of the carrot slices before drying (g).

### 2.11. Vitamin A, Total Phenol Content, and Carotenoid Content Determination

Carotenoids are an important class of natural pigments with vibrant colors and excellent nutritional value [26,27]. In this experiment, the ethanol extraction-colorimetric method was used [28]. First, a sample of 2 g was added to 10 mL of ethanol, heated and stirred, and then cooled and filtered to obtain the filtrate. Next, 5 mL of petroleum ether was added, and the upper layer of petroleum ether was removed after layering and repeated 2–3 times. The extract was placed in a colorimetry dish and the absorbance was measured at a wavelength of 465 nm using a UV-visible spectrophotometer (UV-6000, Beijing General General Instrument Co., Ltd., Beijing, China). A standard curve was drawn and the carotenoid concentration was calculated.

The total phenol test method utilized Folin-phenol colorimetry. Firstly, 1 g of carrot sample was accurately weighed, crushed, and added to 10 mL of distilled water and stirred thoroughly, and filtered, collecting the filtrate. The filtrate was transferred to a 25 mL volumetric bottle, 1 mL of 15% Na_2_CO_3_ solution was added, and distilled water was added to a final volume of 25 mL. The sample solution was placed in the colorimetry dish of the UV-visible spectrophotometer (UV-6000, Beijing General General Instrument Co., Ltd., Beijing, China) with a wavelength of 720 nm. A volume of 2 mL of 0.25 mol/L Folin-phenol reagent was added, mixed quickly, and allowed to stand at room temperature for 30 min. The absorbance of the solution was measured and the measured absorbance value was recorded. The total phenol content in the sample was calculated using the standard curve method based on the absorbance value. The measurement was repeated three times to calculate the average value and obtain the final result.

The measurement of vitamin A content utilized liquid chromatography-mass spectrometry (LC-MS). One gram of dried carrot sample was crushed and screened, then added to 5 mL of methanol for ultrasound extraction for a period of time, filtered, and the filtrate was reserved. Standard preparation: vitamin A standard was dissolved in methanol to prepare different concentrations of standard solutions. The chromatographic conditions: A C18 column was used, with methanol and water as a mobile phase for gradient elution. The detection wavelength was 290 nm. Mass spectrometry conditions: Electrospray ion source (ESI), positive ion mode, monitoring the molecular ion peak of vitamin A. Vitamin A standard solution was injected into the LC-MS, recording the peak area, and preparing the calibration curve. The treated dried carrot sample solution was injected into the LC-MS (UPLC/AB 4000, Waters Company, Milford, MA, USA), which recorded the peak area. The vitamin A content was calculated according to the calibration curve, by calculating the average value and drawing the standard curve.

### 2.12. Infrared Spectroscopy

The dried carrot samples were ground and mixed with potassium bromide (1:100) and milled. The mixed powder was sieved and placed in a press (HY-12, Jiangyin Huayu Pharmaceutical Machinery Co., Ltd., Jiangyin, China) to form a pellet. The sample was then scanned using an FTIR spectrometer (Nicolet iS10, Thermo Fisher Scientific, Waltham, MA, USA) in the spectral range of 400–4000 cm^−1^ with a resolution better than 0.4 cm^−1^.

### 2.13. Scanning Electron Microscopy (SEM)

Carrot dried products were cut into thin slices using a double-sided blade. After drying in a critical point dryer with carbon dioxide, the samples were coated with a conductive adhesive and then sprayed with gold using an ion sputter coater. Scanning electron microscope (SEM) (SU8020, Hitachi High-Tech Corporation, Tokyo, Japan) images were obtained using SEM with an accelerating voltage of 5 kV, a working distance of 7.9 mm, and a magnification of 250×. The same location on various samples was scanned to obtain SEM images.

### 2.14. Low Field NMR

The coil diameter of the nuclear magnetic resonance analyzer (MicroMR12-025V, Niumai Analytical Instrument Co., Ltd., Niumai, Suzhou, China) was 25 mm, and the magnetic field intensity was 12 MHz. The experimental parameters were as follows: the 90-degree pulse width (P1) was 6 μs, the 180-degree pulse width (P2) was 9.84 μs, the repetition sampling waiting time (TW) was 3000 ms, the number of echoes (NECH) was 5000, the number of repetitions (NS) was 4, and the receiver bandwidth (SW) was 200 kHz. Finally, the obtained test data were subjected to 10,000 inversions and plotted using software to obtain the $T_{2}$ spectrum.

### 2.15. Statistical Analysis

All experiments were performed in triplicate, and the results were expressed as mean ± standard deviation. One-way analysis of variance (ANOVA) was performed to determine the differences between groups, with a significance level of 0.05 to determine differences between means.

## 3. Results and Discussion

### 3.1. Dry Characteristics

#### 3.1.1. Drying Rate and Drying Time Analysis

As shown in Figure 2, the drying rate in the control group remained relatively stable, fluctuating within a certain range. Under EHD drying treatment, the drying rate was directly proportional to the voltage within a certain range. This is consistent with the experimental conclusions of Esehaghbeygi et al. [29] on EHD drying of tomatoes and Ding et al. [14] on EHD drying of carrot slices. Increasing the voltage had a significant impact on improving the drying rate, as the drying rate increases faster with higher voltage, discharge frequency, and current amplitude. The stronger the ion wind effect, the faster the loss of moisture. The process of ion wind formation [30] is illustrated in Figure 3. When the needle electrode was charged with a negative voltage, a large amount of positive ions accumulated at the tip of the needle electrode in the needle-plate electrode electric field. Under the influence of the electric field force, electrons moved towards the plate electrode while colliding with molecules in the air, generating more electrons and ions of the same polarity. This drove the directed movement of electrons and ions of the same polarity, resulting in the formation of an ion wind. Furthermore, we found that the drying rates of the two groups with voltages of 38 kV and 44 kV were basically consistent. This indicated that the influence of voltage on the drying rate became very weak when the voltage reached a certain specific value. As shown in Figure 4, EHD drying significantly reduced the drying time. Therefore, the application of EHD drying to carrots was practical.

#### 3.1.2. Moisture Ratio Analysis

Moisture ratio is an important parameter for evaluating the drying process of materials and reflects the degree of material drying. Yang et al. [18] found significant differences in moisture ratio among goji berry samples treated with different voltage values in EHD drying experiments. Compared to the control group, the moisture ratio of goji berry samples dried using EHD was significantly reduced. The higher the voltage, the faster the decrease in moisture ratio [14,20,31]. This conclusion is consistent with the results of our experiment. As shown in Figure 5, the decrease in moisture content in the EHD experimental group was directly proportional to the voltage, and the rate of decrease decreased over time, entering a slow drying phase in the later stages. The moisture content in the control group decreased at a slower rate, and the drying rate remained relatively stable throughout the drying process. We found that the two groups with voltages of 38 kV and 44 kV showed nearly identical moisture content curves. This indicated that the influence of the electric field on moisture gradually saturated after the voltage increased to a certain level. So that even with further voltage increases, the effect on moisture content became very weak.

#### 3.1.3. Analysis of Effective Moisture Diffusivity

Effective moisture diffusivity is influenced by various factors. Paul et al. [32] studied the impact of electrohydrodynamic drying on the drying kinetics, energy consumption, and quality of apple and strawberry slices with thicknesses ranging from 1–4 mm and found that fruit slice thickness significantly affected effective moisture diffusivity and energy consumption, with higher fruit slice thickness leading to decreased drying rates. Based on the experimental data, an lnMR−t diagram was plotted, and a fitting equation was obtained through the least square method, as shown in Table 1. The effective moisture diffusivity was then calculated. As shown in Figure 6, compared to the control group, EHD drying significantly increased the effective moisture diffusivity, which increased with the voltage. The effective moisture diffusivity of the EHD drying group (from low to high voltage) was 5.1, 5.09, 5.1, 7.6, and 8.8 times that of the control group. Higher voltages generated stronger ion winds, which accelerated the loss of moisture. Different letters represent significant differences in mean values (*p* < 0.05).

### 3.2. Quality Characteristic

#### 3.2.1. Color Analysis

Color is one of the key factors affecting product quality and directly impacts sales [33]. Problems such as browning, pigment degradation, and oxidation easily occur during the drying process. As shown in Table 2, carrot slices treated with EHD drying had an increase in brightness by around 20%, a decrease in redness by 18.25%, and a decrease in yellowness by 9.7%. Higher voltages preserved the color of the carrot slices better, and with shorter drying times, oxidation and discoloration can be largely avoided. Overall, the color difference is small, which is consistent with the conclusion drawn by Alemrajabi et al. [34], that the color of EHD dried carrot samples remained almost unchanged. In Bai et al.’s study [35], it was found that EHD drying, as opposed to oven drying, effectively preserved the natural color of seaweed, resulting in an olive-like color, a shinier appearance compared to oven-dried samples, and reduced deformation. This is consistent with our conclusion that, compared to the control group, the EHD drying group had lower color saturation and hue angles in dried carrots due to the decrease in moisture content, resulting in a decrease in the a-value and b-value, and an increase in the hue angle difference. Overall, higher brightness and lower redness and yellowness values make the product more vibrant and of higher quality.

#### 3.2.2. Shrinkage Analysis

As shown in Figure 7, the shrinkage rates for drying at 0, 20, 26, 32, 38, and 44 kV were 74.12%, 83.23%, 75.62%, 77.83%, 75.66%, and 76.11%, respectively. The shrinkage rate of the control samples was slightly lower than that of the EHD drying group. The EHD drying group also exhibited higher smoothness compared to the control group. Different voltage parameters have a lower impact on the shrinkage rate. This was consistent with the findings of Xiao et al. [20] regarding EHD drying of mushrooms and Yang et al.’s research [18] on EHD drying of goji berries, where they noted that different voltages did not significantly affect the shrinkage rate. From the photograph of the dried carrot slices, we could observe that the degree of shrinkage of the secondary phloem and secondary xylem in the carrot was different, which was caused by the internal cellular structure differences. The secondary xylem cells had a higher water content, and they shrunk and wrinkled due to water loss after drying, lacking the support of sieve tubes. The degree of shrinkage in the secondary phloem and secondary xylem was different.

#### 3.2.3. Rehydration Performance Analysis

Rehydration rate is a crucial quality parameter for dried foods. It effectively reflects the degree of internal structural damage. Poor rehydration performance is associated with structural damage and cell shrinkage. In a study on EHD drying of banana slices, Esehaghbeygi et al. [36] found that the samples treated with a voltage of 10 kV exhibited the highest rehydration ability. They attributed this improved performance to the non-thermal nature of EHD drying. EHD drying technology is a non-thermal drying method, and the ion wind only acts on the surface of the material, avoiding damage to the internal structure. As shown in Figure 8, the rehydration performance of the EHD drying group was superior to that of the control group. The influence of different voltage parameters on rehydration performance was not significant. Scanning electron microscopy (SEM) analysis revealed that both the control and EHD-dried samples exhibited uniform tissue structure and large voids, which are important factors affecting rehydration performance [37].

### 3.3. Vitamin A, Total Phenol Content, and Carotenoid Content

The stability of carotenoids and the degree of browning during processing and storage directly affect the quality of the product. Wellnerd et al. [38] studied the formation of Maillard reaction products during carrot heat treatment. EHD drying is a non-thermal drying method that can effectively retain the nutritional components of the material. It has unique advantages in drying thermosensitive substances. Ref. [39] and can avoid the Maillard reaction occurring during effective drying [40]. Carotenoids are a type of plant nutrient that has been well-established to benefit the skin and enhance its defense against UV radiation [41,42]. Vimala et al. [43] studied the retention of carotenoids in sweet potatoes using different drying methods and found that the retention rate was highest in oven drying, followed by frying and sun drying. As shown in Figure 9, EHD drying can significantly retain carotenoids compared to the control group. The carotenoid content in the EHD drying group (voltage from low to high) was 1.87, 1.58, 2.0, 1.60, and 1.94 times that of the control group, respectively. The content of carotenoids increased in a wave-like manner with the increase in voltage. Increasing voltage and shortening drying time had a certain effect on preserving carotenoids.

As shown in Figure 10, the total phenol content in carrots dried by EHD was much lower than that in the control group, with values of 15.73%, 17.18%, 10.83%, 8.38%, and 18.12%, respectively, for voltage levels from low to high. EHD drying had a negative impact on the total phenolic compounds in dried carrot slices. This was because when phenolic compounds were exposed to an electric field, they were polarized and ionized, then decomposed. On the other hand, the production of ozone (oxidant) and air anions during the EHD drying process led to food oxidation. This conclusion is consistent with the study of Ameneh et al. [44] on EHD drying of papaya slices. Humans depend on dietary vitamin A to maintain normal organ structure and function [1]. Moreover, EHD drying also had a negative effect on the vitamin A content in dried carrot slices. As shown in Figure 10, the vitamin A content (voltage from low to high) was 76.49%, 57.25%, 66.02%, 48.09%, and 46.66%, respectively, for the control group. This was because vitamin A was soluble in lipids [45,46], and was oxidized by ozone (oxidant) during the EHD drying process. The higher the voltage, the higher the concentration of ozone produced, and the more significant the oxidation of lipid-soluble vitamin A that occurred. This finding was consistent with the discovery of Alireza et al. [47] when using high-pressure static electricity to thaw frozen tuna.

### 3.4. Infrared Spectroscopic Analysis

The infrared spectrum of carrot slices showed strong absorption bands at 3405 cm^−1^ and 2932 cm^−1^, which corresponded to the characteristic stretching vibrations of O-H and C-H, respectively [20]. The main scaffolding of plant cell walls is formed by cellulose microfibers, which are connected to hemicellulose through hydrogen bonds. This cellulose–hemicellulose network is embedded in the matrix pectin gel, resulting in the stretching vibration of the ester C=O at 1745 cm^−1^ and the symmetric stretching of COO at 1415 cm^−1^, corresponding to carboxylate salts from pectin esters [48,49,50]. Various sugars are present in the pectin gel, and absorption bands at 1078 cm^−1^ and 1139 cm^−1^ indicate arabino and galacto-glucomannans, respectively [51,52,53]. The characteristic protein bands are distributed at 1640 cm^−1^ and 1540 cm^−1^, corresponding to the amide I and amide II groups, respectively [31]. The amide I region is responsible for the stretching vibration of the C=O in peptide bonds, while the amide II region is attributed to the stretching of the C-N bond and bending vibration of the N-H bond in the protein structure. As shown in Figure 11, there were no significant changes in the characteristic peak positions of the control group and the EHD drying group, indicating that the initial functional groups did not undergo noticeable changes. This was consistent with the findings of Xiao et al. [20] on EHD drying of mushrooms and Han et al. [31] on EHD drying of garlic slices, demonstrating that EHD drying can effectively retain the major components of carrots.

Carrots contain about 1.0 g of protein per 100 g edible portion. Protein is a large molecule, and its secondary structure determines its properties [54]. The amide I spectrum analysis was performed within the spectral range of 1600–1700 cm^−1^ for protein analysis. After baseline adjustment, Gaussian deconvolution, second derivative, and curve fitting, the β-sheet was identified in the range of 1600–1642 cm^−1^, followed by the random coil in the range of 1642–1650 cm^−1^, the α-helix in the range of 1650–1660 cm^−1^, the β-turn in the range of 1660–1680 cm^−1^, and the β-antiparallel [55] in the range of 1680–1700 cm^−1^. As shown in Figure 12, the analysis of protein secondary structure in carrot indicated that β-sheet and β-turn structures were the main components of the protein’s secondary structure. Protein secondary structures can be categorized into ordered structures and disordered structures; α-helices and β-sheets represent ordered structures, while β-turns and random coils represent disordered structures. The proportions of disordered structures in the control group and experimental group (from low to high voltage) were 37.9%, 27.3%, 27.7%, 38.0%, 38.5%, and 37%, respectively. As the voltage increased, the proportion of disordered structures in the carrot protein increased. In Soraiyay et al.’s study [56] on EHD drying of egg white, it was found that EHD could cause protein deformation, and the use of foam pads can prevent protein degradation during drying. Singh et al.’s research [57] on EHD drying of wheat indicated that EHD drying can alter protein secondary structure and change hydrogen bonding patterns in the protein. Overall, EHD drying had a significant impact on protein secondary structure, with higher voltages leading to an increase in the proportion of disordered structures. The lower voltage group had a smaller proportion of disordered structures compared to the control group, while the higher voltage group had a similar proportion of disordered structures compared to the naturally air-dried group.

### 3.5. Microstructure Analysis

Shrinkage in fruits and vegetables during drying mainly occurs in the early to middle stages and gradually stabilizes in the later stages. In this experiment, carrots were sliced crosswise, with the center part consisting of secondary xylem and the edge part consisting of secondary phloem. The secondary phloem of carrots contains many conduits for transporting nutrients. The cells in the secondary xylem of carrots are incompletely developed, with fewer organelles, and the intracellular space is almost occupied by vacuoles. The moisture content of the secondary xylem is also higher than that of the secondary phloem [58]. In Wang et al.’s study [59] on low-frequency ultrasound (LFU) pretreatment in pre-irradiation infrared radiation (IW-IR) drying, it was found that the cell structure was severely damaged and cells were visibly ruptured after LFU pretreatment. As shown in Figure 13, the EHD drying group exhibited more pronounced wrinkling compared to the control group, with the secondary phloem retaining its structure and having many pores after EHD drying. In the SEM images of the carrot’s secondary xylem, both the EHD drying group and the control group showed noticeable shrinkage. This is because the cells in the secondary xylem have a high water content, and after drying, they lose water and shrink due to the lack of support from sieve tubes. The cell structures of both parts were relatively intact after EHD drying.

### 3.6. Low-Field NMR Analysis

LF-NMR has the advantages of being simple, fast, and non-destructive, and can reveal the distribution of moisture in fruits and vegetables [60,61]. Similar to previous studies, dried carrot slices showed three peaks. The $T_{21}$ peak has the shortest relaxation time (0.1–10 ms), indicating bound water closely associated with macromolecules. The peak $T_{22}$, with a relaxation time of 10–80 ms, represents immobilized water trapped in the cytoplasm. The $T_{23}$ peak, with the longest relaxation time (80–600 ms), represents free water in vacuoles and intercellular spaces [31,62]. As shown in Figure 14, some samples lacked the $T_{23}$ peak, indicating that free water had been completely removed, and the dried carrot slices mainly contained bound water. The peak position of the spectrum line of the EHD drying group shifted significantly to the left compared to that of the control group, indicating a more significant migration of water from a high degree of freedom to a low degree of freedom during EHD drying, as the ions produced during EHD drying acted on water molecules, increased the polarity of water molecules, and promoted their combination with solids. This finding was consistent with the discovery of Han et al. [31] in EHD drying of garlic. Bound water is an important part of the cell, and EHD drying was able to retain more bound water compared to the control group, better maintaining the cell structure. As shown in Figure 15 and Figure 16, the effect of preserving cellular bound water was best at a voltage of 26 kV, and the effect of preserving cellular immobilized water was best at a voltage of 38 kV, while the preservation of cellular bound and immobilized water was relatively poor at a voltage of 44 kV. When implementing this in practical applications, optimum electric field parameters should be considered.

### 3.7. Statistical Analysis

The Pearson correlation coefficient was calculated through correlation analysis of experimental data, including average drying rate, average drying time, effective moisture diffusivity, rehydration performance, shrinkage, carotenoid content, color difference, color saturation, hue angle, $T_{21}$ peak area, and $T_{22}$ peak area. As shown in Figure 17, a significant correlation was observed between carotenoid content and color difference, and color saturation. Drying time demonstrated significant inverse correlations with effective moisture diffusivity, color difference, rehydration performance, and carotenoid content.

To reflect the influence of different voltages on drying parameters, the data were normalized, and a heatmap was plotted, as shown in Figure 18. It was observed that the preservation effect of carotenoids was the best at a voltage of 32 kV. Saturation and hue angle showed inverse correlations with voltage. The optimal voltage for rehydration performance was 38 kV. Examining the influence of experimental parameters on the quality of dried products can provide a reference for practical applications.

## 4. Conclusions

The EHD drying group demonstrated superior drying efficiency, color, shrinkage, and rehydration performance compared to the control group. EHD drying was able to retain carotenoids to a greater extent than the control group. EHD drying had a negative impact on the total phenolic content and vitamin A content of dried carrot slices. In terms of the analysis of infrared spectroscopy, no significant changes in the characteristic peak positions were observed, indicating that EHD drying could effectively retain the main components of carrots. EHD drying had a significant impact on the secondary structure of proteins, with an increase in voltage leading to an increase in disordered structures. The proportion of disordered structures in the low voltage group was smaller than that in the control group, while the proportion in the high voltage group was almost the same as that in the control group. Under scanning electron microscopy, the secondary xylem of carrots showed severe shrinkage after drying, and the shrinkage phenomenon became more pronounced with increasing voltage. The results of low-field nuclear magnetic resonance showed that EHD drying could retain more bound water compared to the control group, and thus had an advantage in maintaining cell structure.

## Figures and Tables

**Figure 1 foods-12-04228-f001:**
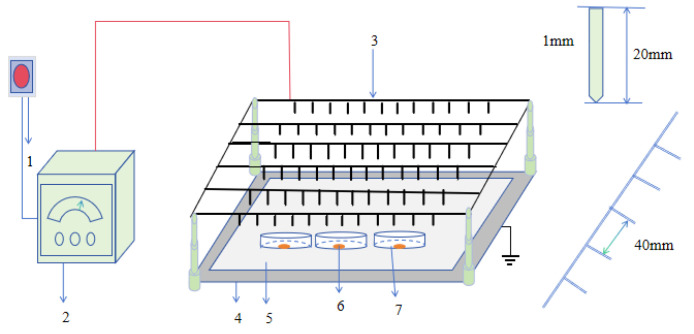
The EHD drying system. 1. electrical source; 2. control system; 3. pin electrode; 4. grounding electrode; 5. dielectric-slab; 6. carrot slices; 7. petri dish.

**Figure 2 foods-12-04228-f002:**
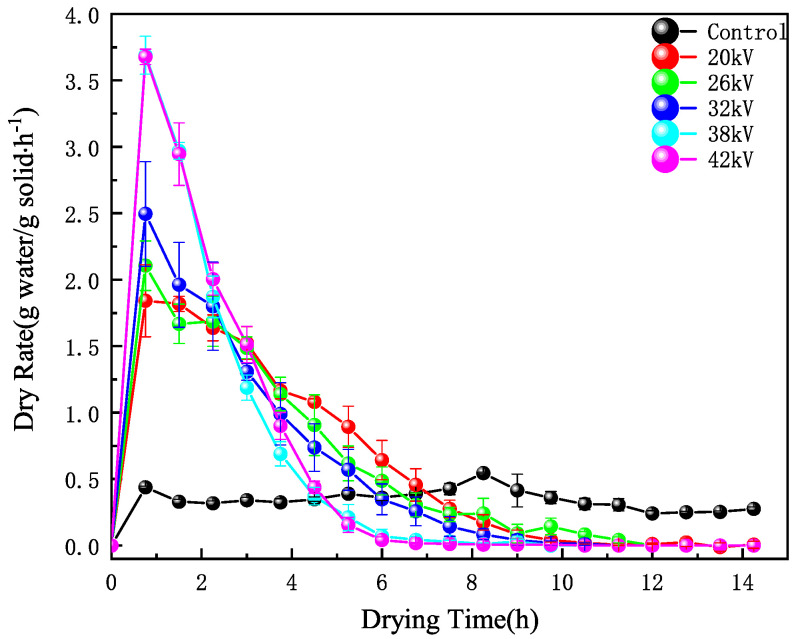
The drying rate over time with different voltages.

**Figure 3 foods-12-04228-f003:**
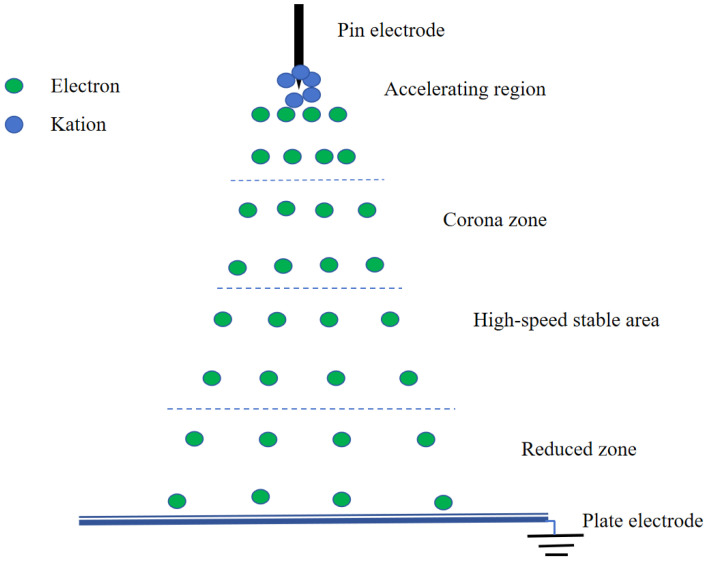
Schematic representation of the particles in the needle-plate electrode.

**Figure 4 foods-12-04228-f004:**
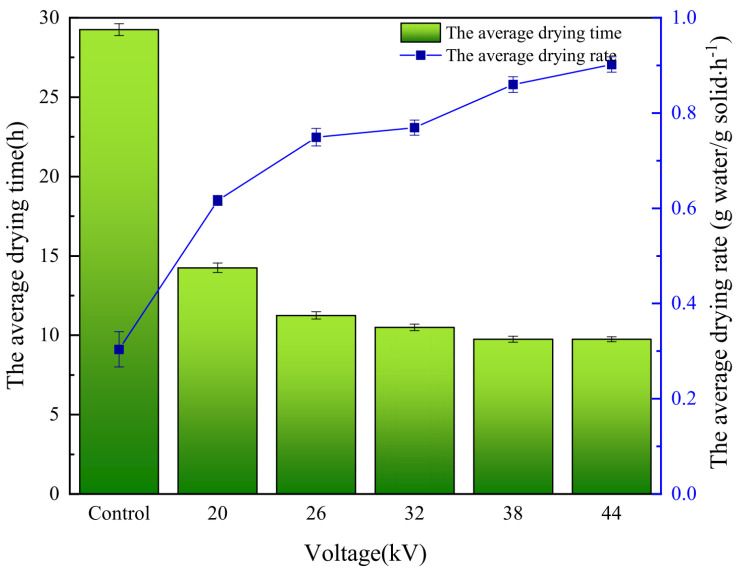
The change curve of the average drying time versus the average drying rate.

**Figure 5 foods-12-04228-f005:**
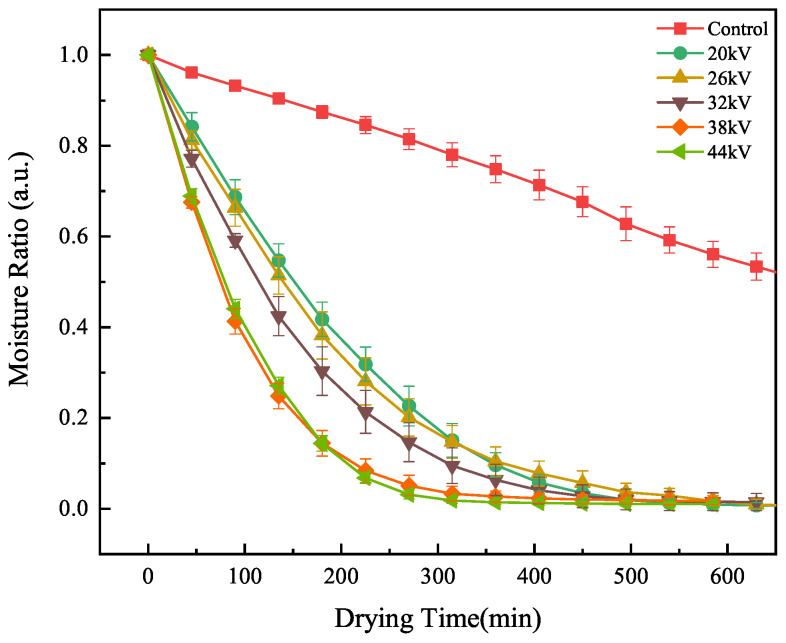
Moisture Ratio of carrots under different voltages.

**Figure 6 foods-12-04228-f006:**
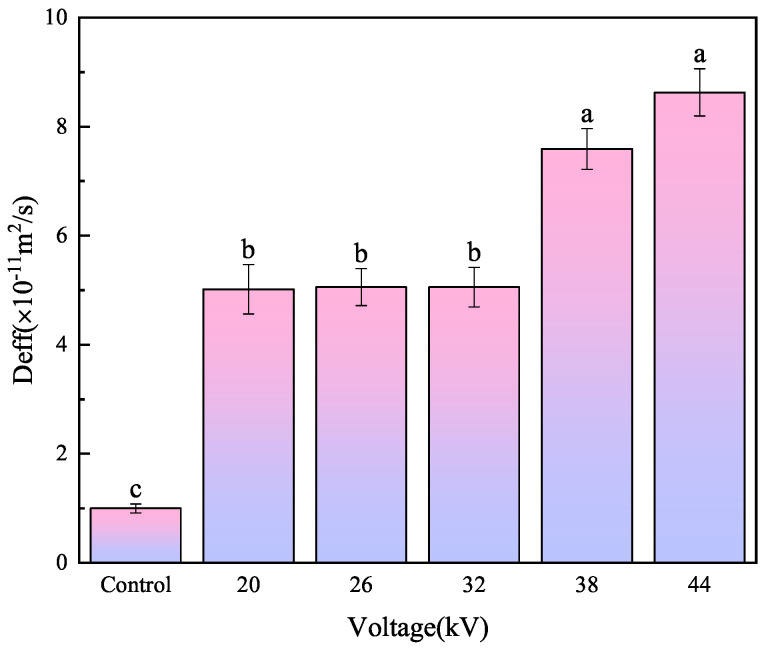
Effective water diffusion coefficient of carrot slices under different voltages. Different letters indicate a significant difference between the sample means (*p* < 0.05).

**Figure 7 foods-12-04228-f007:**
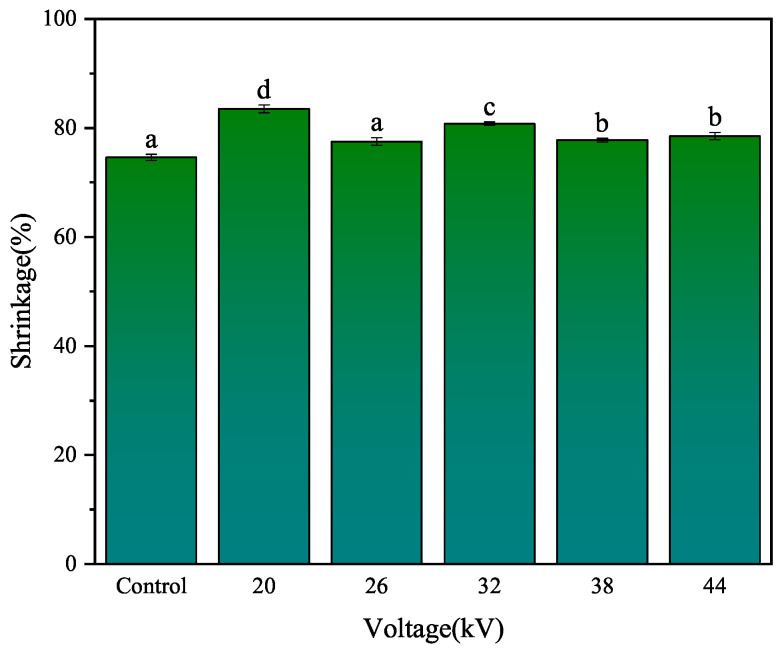
Effect on carrot shrinkage rate with different voltages. Different letters indicate a significant difference between the sample means (*p* < 0.05).

**Figure 8 foods-12-04228-f008:**
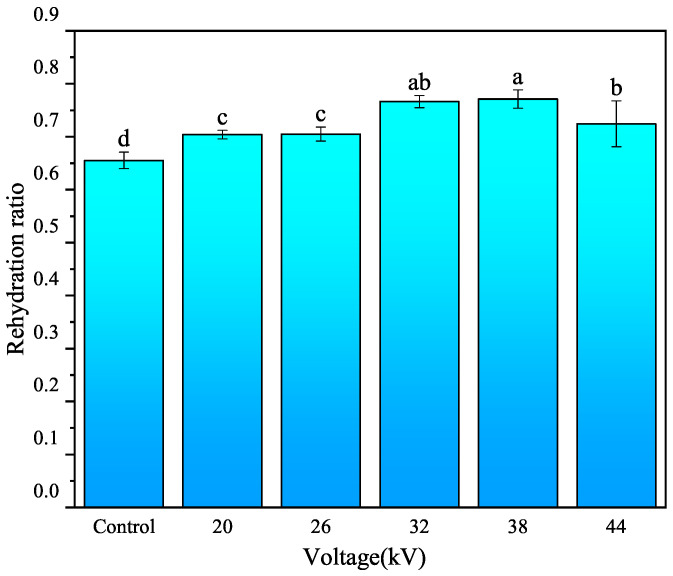
Rehydration properties of carrots after drying with different voltages. Different letters indicate a significant difference between the sample means (*p* < 0.05).

**Figure 9 foods-12-04228-f009:**
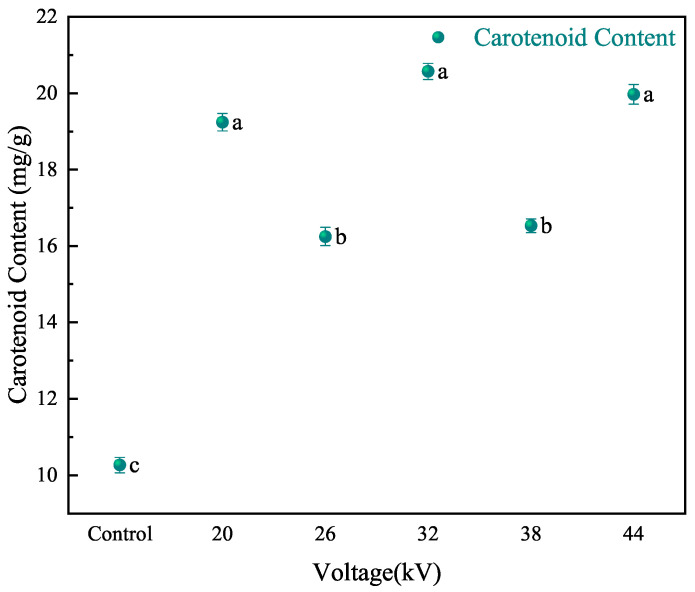
The content of carotenoids in carrots after drying at different voltages. Different letters indicate a significant difference between the sample means (*p* < 0.05).

**Figure 10 foods-12-04228-f010:**
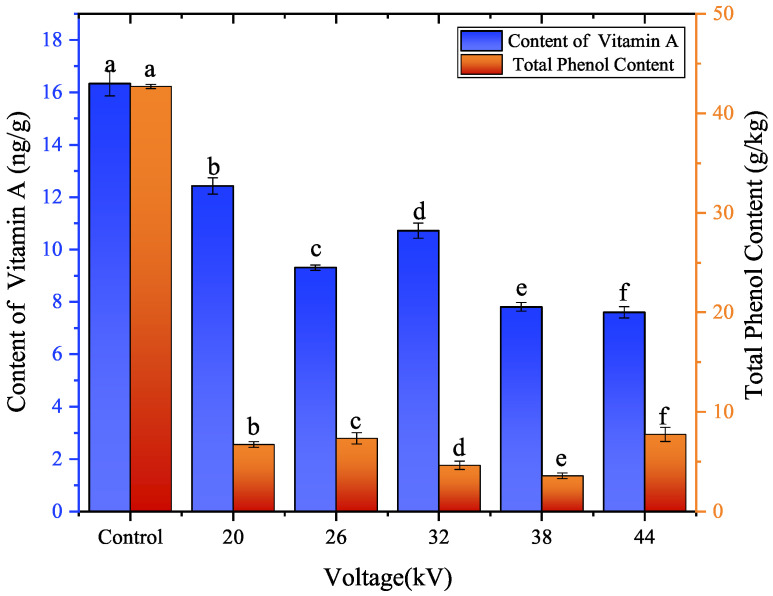
Vitamin A content and total phenol content of carrot slices dried at different voltages. Different letters indicate a significant difference between the sample means (*p* < 0.05).

**Figure 11 foods-12-04228-f011:**
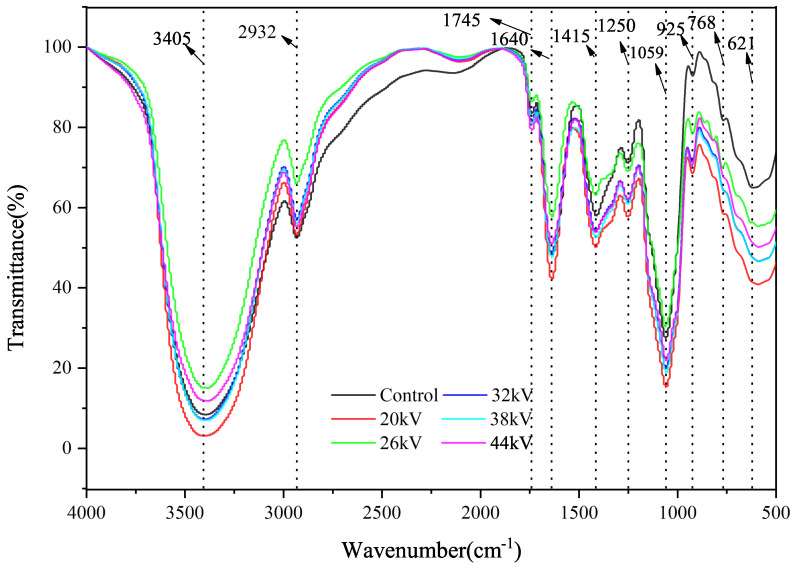
Infrared spectroscopy of carrots after drying at different voltages.

**Figure 12 foods-12-04228-f012:**
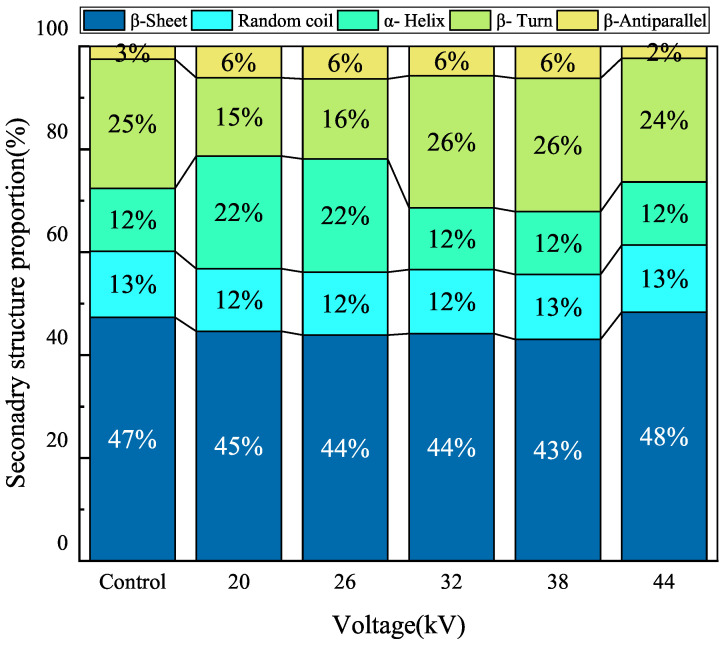
The portion of protein secondary structure of carrots dried with different voltages.

**Figure 13 foods-12-04228-f013:**
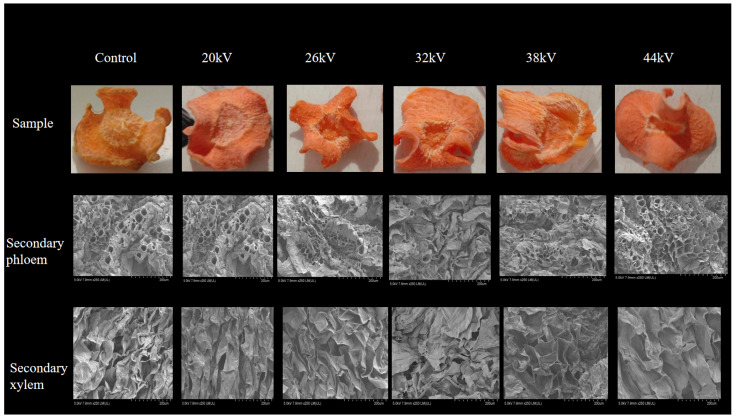
Microstructure of carrot slices’ surfaces after drying at different voltages.

**Figure 14 foods-12-04228-f014:**
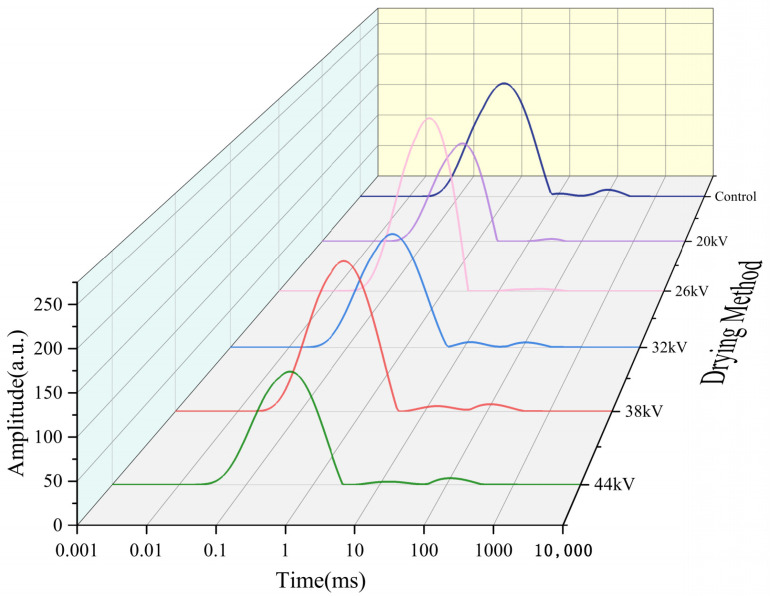
Lateral relaxation time $\left(T_{2}\right)$ spectra of dried carrot slices under different voltages.

**Figure 15 foods-12-04228-f015:**
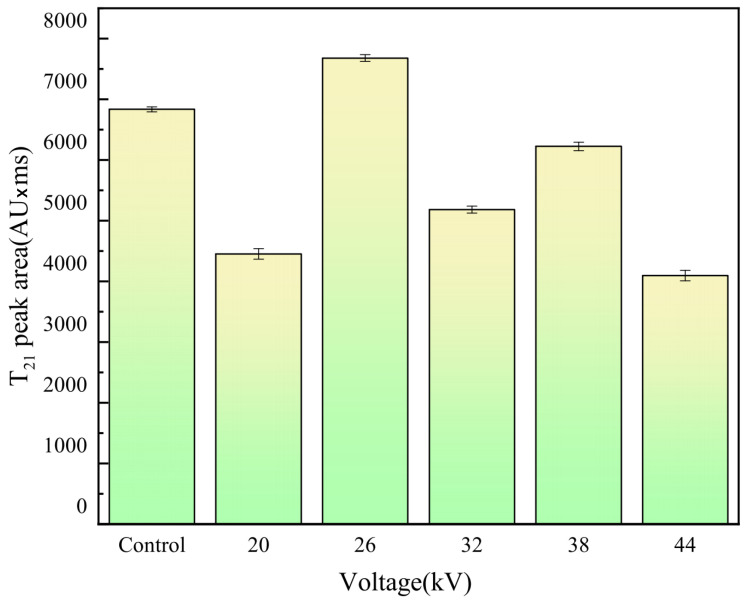
Area difference of T_21_ peaks generated by different voltages.

**Figure 16 foods-12-04228-f016:**
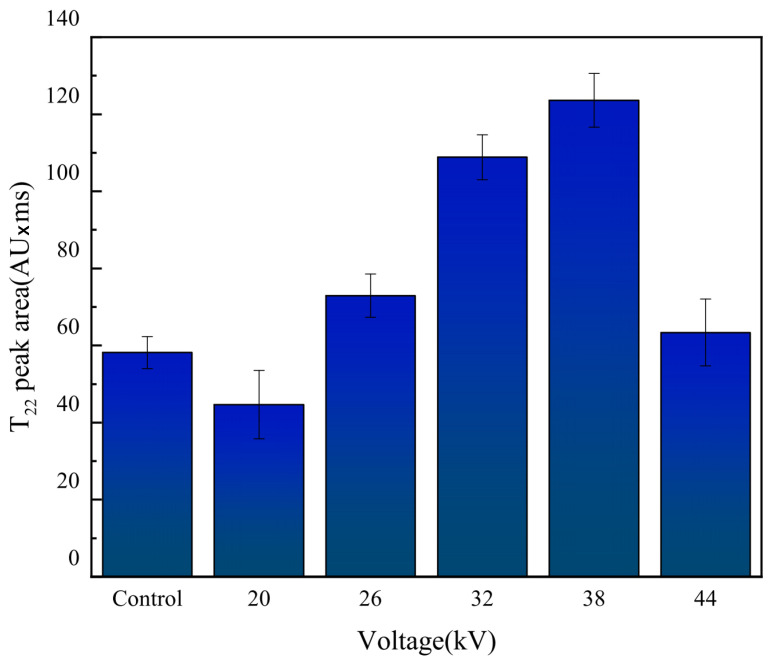
Area difference of T_22_ peaks generated by different voltages.

**Figure 17 foods-12-04228-f017:**
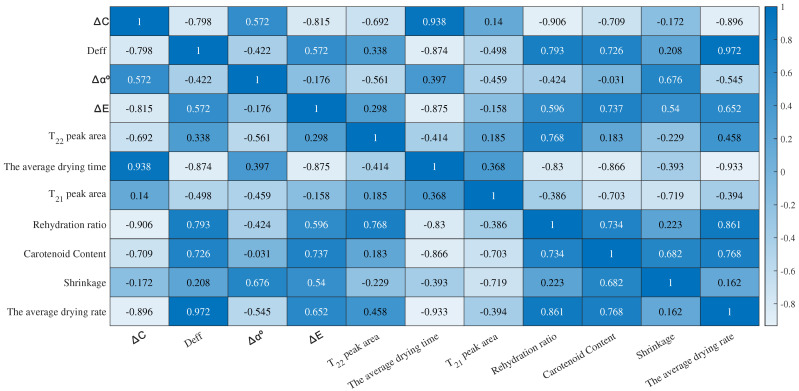
Pearson correlation coefficient matrix.

**Figure 18 foods-12-04228-f018:**
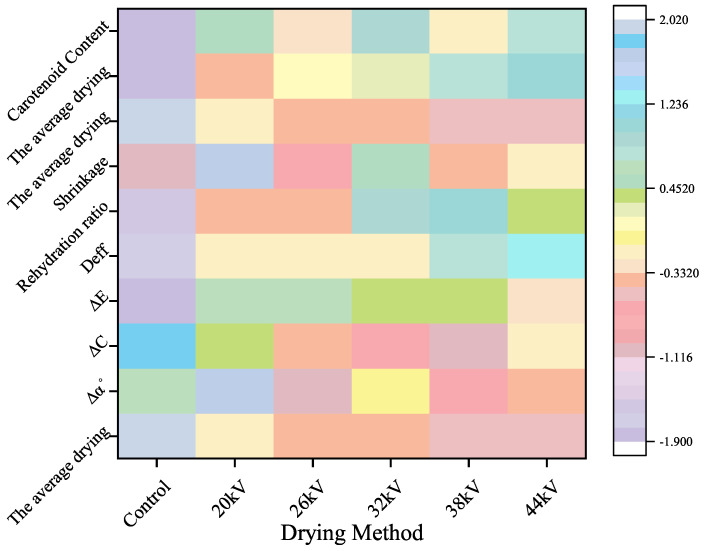
Correlation heatmap of the drying method and drying index.

**Table 1 foods-12-04228-t001:** Table of the parameters of the lnMR−t equation.

Voltage	Linear Model	*R* ^2^	*SSE*	*RMSE*
Control	$-2.463\times 10^{-5}t-0.229$	0.9482	0.8100	0.1567
20 kV	$-1.237\times 10^{-4}t-0.209$	0.9682	2.4420	0.3680
26 kV	$-1.246\times 10^{-4}t-0.309$	0.9865	0.1397	0.1079
32 kV	$-1.247\times 10^{-4}t-0.053$	0.9859	0.4526	0.1866
38 kV	$-1.871\times 10^{-4}t-0.079$	0.9982	0.0132	0.0514
44 kV	$-2.136\times 10^{-4}t-0.218$	0.9812	0.1787	0.1890

**Table 2 foods-12-04228-t002:** Carrot color changes before and after drying.

Voltage	*L* _1_	*a* _1_	*b* _1_	Δ*α*°	Δ*C*	Δ*E*
Fresh	40.330 ± 2.85 ^c^	26.30 ± 2.61 ^a^	32.09 ± 3.72 ^a^	-	-	-
Control	45.230 ± 1.92 ^b^	23.57 ± 4.61 ^ab^	31.7 ± 2.57 ^a^	−0.114 ± 0.30 ^ab^	−2.30 ± 7.35 ab	9.88 ± 3.18 ^b^
20 kV	50.985 ± 3.73 ^a^	19.30 ± 3.42 ^c^	28.76 ± 4.489 ^ab^	0.175 ± 0.15 ^a^	−7.10 ± 5.70 ^bc^	15.38 ± 5.57 ^a^
26 kV	50.277 ± 2.26 ^a^	18.09 ± 2.04 ^c^	26.57 ± 1.54 ^c^	−0.565 ± 0.21 ^c^	−9.50 ± 6.93 ^bc^	15.04 ± 4.50 ^b^
32 kV	50.542 ± 2.57 ^a^	19.80 ± 4.53 ^bc^	27.95 ± 6.84 ^ab^	−0.284 ± 0.26 ^bc^	−10.21 ± 8.04 ^c^	14.69 ± 5.42 ^b^
38 kV	50.450 ± 4.03 ^a^	18.69 ± 1.93 ^c^	28.18 ± 2.16 ^ab^	−0.443 ± 0.32 ^c^	−11.48 ± 3.48 ^c^	14.80 ± 3.97 ^b^
44 kV	50.128 ± 2.22 ^a^	20.51 ± 2.76 ^bc^	31.54 ± 2.68 ^a^	−0.387 ± 0.32 ^bc^	−8.90 ± 5.33 ^a^	13.24 ± 5.50 ^b^

Different letters indicate a significant difference between the sample means (*p* < 0.05). *L*_1_: Brightness value; *a*_1_: red value; *b*_1_: yellow value; Δ*E*: total color difference; Δ*C*: color saturation; Δ*α*°: hue angle.

## Data Availability

Date on this study are available in the article.

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
