# Peer review of "Effect of Electrohydrodynamic Drying on Drying Characteristics and Physicochemical Properties of Carrot"

_foods, 2023, doi:10.3390/foods12234228_

Round 1
Reviewer 1 Report
Comments and Suggestions for Authors
The authors reported their work on the study of EHD on Drying Characteristics and Physicochemical Properties of Carrot. The using method is interesting. I have too many comments for improving readability and emphasizing key results. The manuscript is worth consideration after addressing these comments and revising the manuscript
1. The abstract is overloaded with information. Please exclude non-informative text and export in to the Introduction section.
2. As feedstock for the experiments EHD was used to Influence on Drying Characteristics and Physicochemical Properties of Carrot. One of my initial concerns was whether it would be possible to accurately identify and pinpoint a necessity of EHD in carrot processing. Does the current technologies such as REV or PEF of freeze-drying are not enough according to the product quality of process parameters? Any other challenges? This will make the motivation for this experiment easier to understand in my opinion and introduce the reader better into the topic.
3. Something else missing from the Introduction is a correlation with papers based on a drying pre-treatments: cold plasma, pulsed electric field etc.
4. Line 73: What is high-pressure electric field drying ? The cited literature do not include any information regarding high-pressure electric field drying. It is based on EHD.
5. Additionally, if Ding etc have already published the manuscript based on EHD for carrots. What is principally new in submitted manuscript?
6. M&M. The authors stated that AC voltage range of 0 to 50 kV was used instead of DC source. What is the base for such choose? Most of well known papers based on EHD drying in food application are using the DC source. https://doi.org/10.1016/j.jfoodeng.2021.110611 https://doi.org/10.1016/j.ifset.2021.102859
7. Line 104: The pre-treated carrot slices were placed in the EHD drying system. How pre-treated? Please clarify.
8. R&D overall. Presented results are informative, however I did not find any competitive analysis of quality and drying process parameters with traditional technologies. It would be more interesting to compare suggested ED technology with emerging technologies such as PEF or CP.
Some additional general comments:
1. The overall English of the text can be improved further. In particular, the usage of articles (a, an, the) is very commonly omitted. The authors are invited to make further re-reads and improve the structure of the manuscript. For example, line 60-69. The sentence is 9 lines long.
Based on the above points, I would propose a major revision of the manuscript.
Comments on the Quality of English Language
The authors reported their work on the study of EHD on Drying Characteristics and Physicochemical Properties of Carrot. The using method is interesting. I have too many comments for improving readability and emphasizing key results. The manuscript is worth consideration for publication after addressing these comments and revising the manuscript
1. The abstract is overloaded with information. Please exclude non-informative text and export in to the Introduction section.
2. As feedstock for the experiments EHD was used to Influence on Drying Characteristics and Physicochemical Properties of Carrot. One of my initial concerns was whether it would be possible to accurately identify and pinpoint a necessity of EHD in carrot processing. Does the current technologies such as REV or PEF of freeze-drying are not enough according to the product quality of process parameters? Any other challenges? This will make the motivation for this experiment easier to understand in my opinion and introduce the reader better into the topic.
3. Something else missing from the Introduction is a correlation with papers based on a drying pre-treatments: cold plasma, pulsed electric field etc.
4. Line 73: What is high-pressure electric field drying ? The cited literature do not include any information regarding high-pressure electric field drying. It is based on EHD.
5. Additionally, if Ding etc have already published the manuscript based on EHD for carrots. What is principally new in submitted manuscript?
6. M&M. The authors stated that AC voltage range of 0 to 50 kV was used instead of DC source. What is the base for such choose? Most of well known papers based on EHD drying in food application are using the DC source. https://doi.org/10.1016/j.jfoodeng.2021.110611 https://doi.org/10.1016/j.ifset.2021.102859
7. Line 104: The pre-treated carrot slices were placed in the EHD drying system. How pre-treated? Please clarify.
8. R&D overall. Presented results are informative, however I did not find any competitive analysis of quality and drying process parameters with traditional technologies. It would be more interesting to compare suggested ED technology with emerging technologies such as PEF or CP.
Some additional general comments:
1. The overall English of the text can be improved further. In particular, the usage of articles (a, an, the) is very commonly omitted. The authors are invited to make further re-reads and improve the structure of the manuscript. For example, line 60-69. The sentence is 9 lines long.
Based on the above points, I would propose a major revision of the manuscript.
Author Response
After making the revisions, we have gained a deeper understanding of the question you raised, and our response has been submitted as an attachment.

Reviewer 2 Report
Comments and Suggestions for Authors
The paper entitled “Effect of electrohydrodynamic drying on drying characteristics and physicochemical properties of carrot” investigates the possibility of using the electrohydrodynamic drying technology to overcome the main disadvantages of obtaining dried carrots by natural drying.
The authors will find bellow some minor corrections and adjustments that should be addressed.
- There are some phrases rather difficult to understand (e.g. „Electrohydrodynamic drying technology has advantages such as high efficiency, low energy consumption, and non-thermal drying technology [11–13], and is gradually becoming a new research hotspot in the drying industry” „The needle electrodes were connected to the high-voltage power supply control system. The needle electrodes were made of stainless steel and connected to the high-voltage power control system.” etc.). It is recommended to revise them.
- At the end of the Introduction section there is a phrase about preliminary research conducted on the application of electrohydrodynamic drying technology for carrots. Are these preliminary researches mentioned from literature or they were carried out for the present work. With what results?
- In „Materials and Methods” section, the provider and the provenience country should be added for each apparatus used in the experimental program.
- From subsection 2.2 (line 98), one can understand that there were only two needles used since the distance between them is of 40 mm. The expression must be revised.
- In subsection 2.9, information about the way of establishing the volume should be included.
- A brief explanation for the chosen temperature for rehydration tests should be added in section 2.10.
- In subsection 3.3, an accurate explanation about the similarity of the results recorded for 20, 32, 44 voltages and for 26 and 38 voltages should be provided. From figure 10, it seems that a treatment of 20 kV has the same impact on the carotenoids content as the one applied at 44 kV for example.
- Apart the carotenoid content, were tests conducted on different other constituents in order to study the impact of the used technology on their content? With what results?
Comments on the Quality of English LanguageMinor editing of English language required
Author Response

(The authors gave the same response as above.)

Reviewer 3 Report
Comments and Suggestions for Authors
Effect of Electrohydrodynamic Drying on Drying Characteristics and Physicochemical Properties of Carrot
Comments:
The natural drying method is sun drying. Authors used the drying method without source of heat and without controlled air flow and very low temperature of drying. Application of this method is uneconomical. I suggest to not use the term naturally dried method but describe this sample as variant without application of voltage kV. The control sample should be defined with chapter material and methods. What was the humidity of air.
Figure 5 is typo (“dyring”)
Figure 6: How Authors describe the same course of moisture ratio curve obtained for 38 and 44 kV.
What was the final water content in samples
Whether the equilibrium moisture content for the control sample has been achieved.
The first sentences of conclusions was obvious.
In conclusion Authors should indicate the best conditions of treatment for practical application. In my opinion, the authors should have explained the effect of high or low voltage on specific characteristics rather than referring to a control sample.
Comments on the Quality of English Language
Please correct the typos.
Author Response

(The authors gave the same response as above.)

Reviewer 4 Report
Comments and Suggestions for Authors
The authors in this manuscript investigate influence of electrohydrodynamic drying method on selected quality parameters of carrot slices.
Unfortunately in my opinion, the assumed methodology did not allow reliable assessment of influence of drying process. You used just 3 slices of carrots. What was the mass of each samples? Plant material could varied and you could not ensure uniformity of samples. Did you make any replication? Core and cortical part of carrot roots have different properties. Different proportion of these parts, also strongly affected on obtained result (such as carotenoid content).
in my opinion, although the rest of the paper is quite well written, due to the insufficient sample size, no reliable conclusions can be drawn from it.
Other comments are presented below
In introduction you presented advantages of different drying methods. But firstly you presented methods studied by Mohammadi et. al., then you move to influence of blanching, than you again back to drying methods. It sounds a little chaotic for me.
Line 32 Bifidobacteria? In my opinion all live cells contain nucleic acid.
Line 32-36 These statements sound very strong for me. Does carotenoids prevent respiratory infection or just lowering the risk? If yes, provide medical evidences.
Line 38 Maybe add few words that carrot could be preserved by drying.
Line 39 Does other drying methods requires less materials and manpower?
Line 45 What other products were radiation energy dried?
Line 70 In my opinion firstly provide brief description, what electrohydrodynamic drying is.
Line 71 Is it non-thermal drying?
Line 74 Can accelerate drying rate or what?
Line 88 Carrot roots were cut into slices or slices were cut from core part of carrot?
Line 105 Did you use just three slices for each experiment? How did you ensure homogeneity and repeatability of your experiment?
Line 106 Control sample is natural dried or WHD with 0 kV?
Line 113 What is ms?
Line 121 Expand abbreviations MR and Mt?
Line 138 L is rather lightness. What color space you used Lab or L*a*b*?
Line 145-148 How did you measure volume of samples?
Line 156-166 Provide more details, e.g. sample mass, solvent volume.
Author Response

(The authors gave the same response as above.)

Round 2
Reviewer 1 Report
Comments and Suggestions for Authors
The authors answered my questions in current manner. The manuscript now can be accepted.
Comments on the Quality of English LanguageModerate editing of English language is required.
Author Response
Thank you very much for your suggestion. Our response is submitted as an attachment.

Reviewer 3 Report
Comments and Suggestions for Authors
The paper was improved and can be accepted.
Author Response

(The authors gave the same response as above.)

Reviewer 4 Report
Comments and Suggestions for Authors
The authors have revised the manuscript, improved it and incorporated most of my suggestions. Some of the methodologies used still raise my doubts.
Line 1, "the study explored mechanism of the aplication" - this sentence is confusing, please rewrite it.
Line 27, Carrot contains low amount of ascorbic acid.
Line 155, Delta is diffirence, Why you used it in case of color saturation and hue angle.
Line 192, I think methanol solution is better extraction solvent for phenolics then distilled water.
Line 207-208 What was reference for that? Carotenoids are not disolve in water. Maximum absorbance of carotenoids is abaut 450 nm. Why you used 290 nm?
Line 368, figure 9, Do not connect points by lines.
Line 469-473 Ozon production, polarization and ionization during EHD drying are just speculations. After all, ozone concentrations have not been analysed.
Author Response

(The authors gave the same response as above.)
